# High-Resistant Starch Based on Amylopectin Cluster via Extrusion: From the Perspective of Chain-Length Distribution and Structural Formation

**DOI:** 10.3390/foods13162532

**Published:** 2024-08-14

**Authors:** Wen Ma, Junyu Tang, Huan Cheng, Jinhu Tian, Zhengzong Wu, Jianwei Zhou, Enbo Xu, Jianchu Chen

**Affiliations:** 1National Engineering Laboratory of Intelligent Food Technology and Equipment, Zhejiang Key Laboratory for Agro-Food Processing, Fuli Institute of Food Science, College of Biosystems Engineering and Food Science, Zhejiang University, Hangzhou 310058, China; 22213083@zju.edu.cn (W.M.); tangjunyu@zju.edu.cn (J.T.); huancheng@zju.edu.cn (H.C.); jinhutian@126.com (J.T.); 2Innovation Center of Yangtze River Delta, Zhejiang University, Jiaxing 314102, China; hizjw@163.com; 3State Key Laboratory of Biobased Material and Green Papermaking, School of Food Science and Engineering, Qilu University of Technology, Shandong Academy of Sciences, Jinan 250353, China; wzzqlu@hotmail.com

**Keywords:** resistant starch type-III (RS3), extrusion, crystal structure, chain-length distribution, retrogradation

## Abstract

Resistant starch (RS) has the advantage of reshaping gut microbiota for human metabolism and health, like glycemic control, weight loss, etc. Among them, RS3 prepared from pure starch is green and safe, but it is hard to achieve structural control. Here, we regulate the crystal structure of starch with different chain-length distributions (CLDs) via extrusion at low/high shearing levels. The change in CLDs in extruded starch was obtained, and their effects on the fine structure (*Dm*, *d_Bragg_*, *d_Lorentz_*, degree of order and double helix, degree of crystal) of RS and its physicochemical properties were investigated by SAXS, FTIR, XRD and ^13^C NMR analyses. The results showed that the RS content under a 250 r/min extrusion condition was the highest at 61.52%. Furthermore, the crystalline system induced by high amylopectin (amylose ≤ 4.78%) and a small amount of amylose (amylose ≥ 27.97%) was favorable for obtaining a high content of RS3-modified products under the extruding environment. The control of the moderate proportion of the A chains (*DP* 6–12) in the starch matrix was beneficial to the formation of RS.

## 1. Introduction

Resistant starch (RS) is a new dietary fiber, and this concept derived from starches or starch-based materials was first proposed by Englyst et al. [1]. Commonly, RS is seldom absorbed by the small intestine but is utilized by the colorectal flora, thus regulating intestinal health and human metabolism, stabilizing blood glucose and controlling body weight [2,3]. RS is usually divided into five types (RS1-5), and among them, RS3 is a retrograded starch, which is an anti-digestion structure made of pure starch and is mostly prepared and used in green, safe and sustainable modes such as retrogradation [4,5,6].

The digestibility of pure starch is closely related to its size and crystal structure, naturally or artificially. It is composed of two basic types of chains in the presence of (1) linear (or low-branched) amylose and (2) highly branched amylopectin [7]. The amylopectin cluster (APC) growth model has been regarded as a backbone model based on the amylopectin unit proposed by Bertoft et al. [8]. The chain structure (or called chain-length distribution, CLD) of the APC model can be divided into the A chain (outer chain, with a degree of polymerization (*DP*) between 6 and 12), B chain (inner chain: B1 chain, 13 < *DP* < 24; B2 chain, 25 < *DP* < 36; and B3 chain, *DP* > 36) and C chain (main chain) [9]. Different treatments can affect the crystalline behaviors of diverse CLDs, thereby altering the formation and arrangement of starch structures such as crystal form and lamellar thickness, further affecting physicochemical properties such as digestibility, viscosity and thermal stability [10]. For example, Zhang et al. [11] investigated the relationship between the SDS of pasted maize starch and its molecular fine structure and found that there is a parabolic relationship between SDS content and the weight ratio of amylopectin short chains to long chains. A too low or high ratio of A to B chains may inhibit the digestive rate by promoting regrowth and the inherent high branching density of A chains, respectively.

Starch structure is often modified by using an extruder with the coupling of multiple physical actions such as heating, mechanical shearing and pressing [12,13]. In contrast to the conventional high-liquid processing system, where starch chains are spatially dispersed, the shear action of high-solid extrusion induces starch chain orientation and restricts disordered expansion, which can reconfigure the starch CLD regeneration [14]. It is important to investigate the effect of the extrusion shear field (represented by screw regulation) on the CLDs of starch for the reformed structure of the RS product. In recent years, many studies have been conducted on the efficient preparation of RS by extrusion processing. Gulzar et al. [15] extruded rice flour by adjusting screw speed, feed water and barrel temperature, and its RS content increased from 3.0% to 6.2%. Zeng et al. [16] co-extruded chlorogenic acid with rice starch and found that SDS and RS contents increased to 18.45% and 23.81%. However, the structural change and formation mechanism of RS from pure starch induced by differential extrusion shearing are still unknown.

Therefore, this study aims to investigate the changes in the structure and physicochemical properties of extruded starch with regenerated crystals from pure starch with different CLDs (controlled with a low ratio of amylose to amylopectin, i.e., amylose content ≤ ~30%), and focuses on the effects of screw speed on the CLD modification and structural formation of RS.

## 2. Materials and Methods

### 2.1. Materials

Normal maize starch (NS, 99% purity, amylose content ~30%) and waxy maize starch (WS, 99.3% purity, fat content ~0.28%) were obtained from Datang Biological Engineering Co., Ltd. (Hefei, China) and Baolingbao Biological Co., Ltd. (Dezhou, China), respectively. Porcine pancreatic enzyme (1000 U/mL) and invertase (300 U/mg) were supplied by Sigma-Aldrich Chemical Co. (Milwaukee, WI, USA). Glucosidase (800 U/mL) was bought from Aladdin Bio-Chem Technology Co., Ltd. (Shanghai, China). The Megazyme glucose assay kit was purchased from Megazyme International Ireland Ltd. (Wicklow, Ireland).

### 2.2. Extrusion Sample Preparation

WS (Control_1_) and NS (Control_2_) were compounded in fixed ratios of 1:6, 1:2, 1:1, 2:1 and 6:1 (i.e., group labels) as the proportion of amylopectin decreased. The moisture content of the starch was adjusted to 35% before extrusion with a twin-screw extruder (TES 27 EBYJ, Guangzhou, China). Two levels of screw speeds, high (250 rpm, labeled as HS) and low (50 rpm, labeled as LS), were set to provide different shear strengths. The temperatures in the five-barrel zones from the feeder to the die zone were maintained at 40, 55, 70, 85 and 100 °C, respectively. After being cooled and regenerated at 4 °C for 48 h, the starch samples were freeze-dried. Each dried sample was placed in a grinding jar and ground with a grinder (Tube Mill 100 control, IKA GmbH, Staufen, Germany) at 7500 rpm for 27 s. The samples were then poured out and passed through several sieves (10 U.S. mesh, 2000 μm, 20 U.S. mesh, 850 μm and 200 U.S. mesh, 74 μm) to obtain starch granules of different particle sizes for different analyses.

### 2.3. In Vitro Digestive Simulation

Rapidly digestible starch (RDS), slowly digestible starch (SDS) and RS contents were determined according to the method of Englyst et al. [17] with minor modifications. The starch samples were passed through a 20 U.S. mesh to a 10 U.S. mesh sieve, which corresponds to the particle size range of the chyme from gastric emptying to the small intestine and possibly minimizes the degradation of the crystal structure induced by mechanical milling [18].

A total of 600 mg of samples was weighed into 10 mL of 0.25 M sodium acetate buffer with 5 mL of an enzyme mixture in which the ratio of porcine pancreatic enzyme, glucosidase and invertase enzyme activities was 5:7:4 (×10^2^ U/mL). The mixture was shaken at 37 °C in a thermostatic water bath, and 0.5 mL of the digest was transferred to 20 mL of ethanol (66%, *v*/*v*) solution to inactivate the enzyme at 0, 20 and 120 min (recorded as FSG, G20 and G120, respectively). A total of 10 mL of 7 M potassium hydroxide solution was added to the 120 min digest, and the slurry was then bathed in ice water for 30 min to hydrolyze the starch granules. Then, 1 mL of the hydrolysate was added to 10 mL of 0.5 M sodium acetate buffer, and 50 μL of amyloglucosidase was added. After shaking at 70 °C for 30 min, the slurry was diluted with deionized water to assay total glucose (TG). Glucose content was determined using a Megazyme glucose assay kit (GOPOD method), and the contents of RDS, SDS and RS were calculated using the following equations:(1)$$RDS(\%)=\frac{G20-FSG}{TS}\times 0.9\times 100$$
(2)$$SDS(\%)=\frac{G120-G20}{TS}\times 0.9\times 100$$
(3)$$RS(\%)=\frac{TG-G120}{TS}\times 0.9\times 100$$
where the starch fraction values are expressed as polysaccharides, and a factor of 0.9 was used to convert the measured glucose values to starch. FSG is the free-sugar glucose, G20 and G120 represent the glucose of 20 min and 120 min respectively, and TS is the total starch content.

### 2.4. CLD Determination

#### 2.4.1. Size-Exclusion Chromatography (SEC)

Amylose was detected by debranching with reference to the method of Wu et al. [19] using an LC-20AD SEC system (Shimadzu, Japan) coupled with combination columns of GRAM 1000 and GRAM 100 (Pss, Mainz, Germany) and a refractive index detector of RID-10A (Shimadzu, Japan). The mobile phase was DMSO/LiBr (5%) at a flow rate of 0.6 mL/min. Then, 1 mL of 0.1 M acetic acid buffer (10%, *w*/*w*) and 5 μL NaN_3_ (1 g/mL) were added to 4 mg of starch, and the sample was gelatinized for 15 min in a boiling water bath and then cooled. Next, 2.25 μL of pullulanase (45 U/g) and 6.4 μL of isoamylase (32 U/g) were added to the sample, and the sample was shaken at 37 °C for 3 h and then placed in a boiling water bath for 15 min to inactivate the enzyme. The sample was freeze-dried and thoroughly dissolved in 1 mL of DMSO (containing 0.5% LiBr) (4 mg/mL). A total of 1 mL of solution was drawn from the injector for detection. The flow rate was 0.6 mL/min, and the column temperature was 80 °C. A series of pullulanase standards with molecular weights ranging from 342 to 2.35 × 10^6^ Da was used to establish the calibration curve. The amylose content (*AC*) and branching degree (*BD*) were determined using the following equations [20]:(4)$$AC\left(\%\right)=\frac{{AUC}_{DP100∼20,000}}{{AUC}_{DP1∼100}+{AUC}_{DP100∼20,000}}\times 100$$
(5)$$BD\left(\%\right)=\frac{1}{N_{de}(X)}$$
where AC is calculated by dividing the area under the curve (AUC) of *DP* 100−20,000 of the SEC weight CLDs by the area under the curve of DP 1−20,000, and N_de_(X) represents the chain number distribution.

#### 2.4.2. Fluorophore-Assisted Capillary Electrophoresis (FACE)

The FACE technique was used to detect amylopectin. ~0.3 mg of enzymatically debranched lyophilized starch in a centrifuge tube and label the starch with APTS. The MDQ + Face system was used, with a solid-state laser-induced fluorescence detector and an argon ion laser as excitation sources and a carbohydrate separator (Beckman Coulter) as buffer. The number of CLDs of amylopectin, N_de_(X), was determined by placing the samples in N-CHO-coated capillary tubes at a temperature of 25 °C and an assay voltage of 30 kV [21].

### 2.5. Long-Range Ordering Determination

#### 2.5.1. X-ray Diffraction (XRD)

XRD was used to analyze the crystal form and relative crystallinity (RC) of starch. About 50 mg of ground and sieved (through a 200 U.S. mesh, 74 μm) sample was scanned using a diffractometer (Bruker D8 Advance, Bruker Ltd., Berlin, Germany), operated at 40 kV and 4 mA, using Cu-Kα radiation (λ = 0.1542 nm). The sample was scanned from 4° to 40° (2θ scale) with a step size of 0.02° and a speed of 2°/min. The diffraction peaks of the crystalline region and the total diffraction peaks were analyzed by MDI-Jade 6 software, and the RC was calculated by the following equation:(6)$$RC\left(\%\right)=\frac{S_{diffraction\;peaks\;of\;crystalline\;region}}{S_{total\;diffraction\;peaks}}\times 100$$

#### 2.5.2. Small-Angle X-ray Scattering (SAXS)

The lamellar structure of starch was determined by Zhang et al. [22] using a SAXS system (Bruker SAXS GmbH, Karlsruhe, Germany) with some modifications. Starch samples were immersed in deionized water to keep them in a glassy nematic state. Each sample was loaded into a detection cell, with the voltage and current set to 40 kV and 50 mA, respectively, and irradiated for 5 min using a SAXS system equipped with an X-ray generator and Cu-Ka radiation (*λ* = 0.1542 nm). Data were collected in the range of scattering vectors (q) from 0.007 to 0.226 Å^−1^. Based on the relationship between the relative peak intensity (I) and the scattering vector (q), the original long period d_Bragg_ of the natural starch arrangement was calculated, and based on the Lorentz correction method, a plot of θ^2^I(θ) versus θ was obtained to calculate the d_Lorentz_ of the newly formed crystalline lamellae. The equations involved in the calculations are as follows, where 2θ is the scattering angle, and *λ* is the scattering wavelength:(7)$$q=\frac{4\pi \sin\theta }{\lambda }$$
(8)$$d_{Bragg}=\frac{2\pi }{q}$$

### 2.6. Short-Range Ordering and Basic C Skeleton Determination

#### Fourier Transform Infrared Spectroscopy (FTIR)

Starch samples were mixed with dried KBr in the ratio of 1:100 and then pressed into a thin flake after thorough grinding. The pressed semi-transparent slices were placed in an FTIR instrument with a scanning range of 4000–400 cm^−1^ and a resolution of 4 cm^−1^. Spectra with a wave number range of 1200–800 cm^−1^ were selected for deconvolution to obtain the absorption peak height at 1047 cm^−1^, 1022 cm^−1^ and 995 cm^−1^. The degree of order (*DO*) R_1047/1022_ and the degree of double helix (*DD*) R_995/1022_ were also calculated to characterize the short-range order of the starch samples [23].

### 2.7. ^13^C-Nuclear Magnetic Resonance (^13^C NMR)

Approximately 50 mg of each sample (sieved through 200 U.S. mesh, 74 μm) was weighed and detected using ^13^C nuclear magnetic resonance (^13^C-NMR) fitted with a dual resonance H/X CP-MAS 4 mm probe. The resonance frequency was set to 100.62 MHz, the spectral width to 4 kHz, and at least 1600 scans were used to acquire each spectrum. The cross-polarization contact time was 1.8 ms, and the cyclic delay was 2 s [24]. PeakFit 4.12 software was used to calculate the relative content (%) of the carbon chemical shift regions C1, C4 and C6 of the starch samples, 96–106 ppm for C1, 79–83 ppm for C4 and 57–66 ppm for C6 [25].

### 2.8. Scanning Electron Microscope (SEM)

A small amount of samples (sieved through 200 U.S. mesh, 74 μm) were uniformly coated with conductive adhesive and sprayed with gold. The morphologies of the RS samples were observed by SEM (GeminiSEM 300, Carl Zeiss AG, Oberkochen, Germany) at an accelerating voltage of 3 kV at ×10,000 magnification [6].

### 2.9. Differential Scanning Calorimetry (DSC)

The thermal properties of starch samples were determined by using DSC from Mettler-Toledo (Zurich, Switzerland). The dry samples (about 3 mg) were mixed with deionized water in a sealed crucible at a fixed ratio (1:3, *w*/*v*) and equilibrated at room temperature for at least 12 h. The DSC scanning temperature range was 20–110 °C, and the heating rate was 10 °C/min. An empty crucible was used as a control for the assay process. The changes in enthalpy (Δ*H* in J/g of dry starch) for gelatinization, onset temperature (*T_o_*), peak temperature (*T_p_*) and conclusion temperature (*T_c_*) were calculated using TA Instruments Inc. [26].

### 2.10. Rapid Visco Analyzer (RVA)

RVA (TecMaster, Perten Instrument Co., Ltd., Stockholm, Sweden) was used to determine the pasting properties of starch. A suspension was prepared by weighing 3 g of dry samples and adding 25 g deionized water. Measurement procedure: The agitation speed was set at 960 rpm for the first 10 s and then kept at 160 rpm. The temperature was increased from 50 °C to 95 °C at an increasing rate of 5 °C/min and held at 95 °C for 5 min, and then cooled down to 50 °C at a cooling rate of 6 °C/min and held for 2 min. Peak viscosity (*PV*), trough viscosity (*TV*), final viscosity (*FV*), breakdown value (*BD*), setback value (*SB*), pasting temperature (*P_temp_*_._) and pasting time (*P_time_*) were obtained from pasting viscosity curves [27].

### 2.11. Statistical Analysis

The results of repeated experiments were expressed as the mean ± standard deviation (SD). Data between multiple groups were analyzed by a one-way ANOVA using IBM SPSS 23.0 software, and significance was analyzed by a Duncan’s test (*p* < 0.05).

## 3. Results and Discussion

### 3.1. RS Content of Extruded Starch

The simulated in vitro digestibility of extruded starch was measured according to the sequential digestion patterns of RDS, SDS and RS, with their percentages shown in Figure 1. The results of NS, cooked NS (cooked in boiling water and then cooled to regenerate, labeled as CNS), WS and cooked WS (labeled as CWS) were also placed to contrast the effect of extrusion. Compared with raw starch, the RS content of cooked starch decreases, while extrusion can increase the RS content to a greater extent. Except for LS_2:1_, all other samples showed a high RS content (between 40.77 and 61.52%) under the extrusion with a low/high shearing process, indicating an effective RS3 preparation through extrusion. Most of the RS content of extruded starch at the high level of screw shearing was significantly higher than that at the low shearing condition. Li et al. [27] reported that starch digestion involves the loss of granular structure, growth rings, chain cluster and supramolecular structure, and it is presumed that starch is melted and depolymerized to a great extent at a high screw speed. It may facilitate the regeneration of new resistant structures (details in Section 3.3 and Section 3.4).

In addition, as the content of NS in the extruded starch increased, the content of RS at both high and low shearing levels showed a “U-shaped” trend. That is, groups labeled as Control1, 1:6, 6:1 and Control2 all exhibited a high conversion of RS (with different structures and morphologies, see below), while the change in the content of SDS was somewhat negatively correlated with it (r = −0.024, *p* < 0.05). Among them, the pure amylopectin group via extrusion at the high shearing level had the highest RS content (i.e., HS_Control1_: 61.52%). It is speculated that in the extrusion system, the high shear forces generated by the screw rotation shears the branched side branches and the backbone of the main chains of starch [28], resulting in a large number of short-branched or amylose chains that are highly flexible. It may promote starch intermolecular bonding during the growth and formation of new helical clusters [29]. It is inferred that a similar principle applies to the amylose induction system (HS_6:1_: 27.97%), and the above specific DP classification will be discussed in detail in Section 3.2. Based on the results of the digestion test, eight groups of high-RS samples were selected for further analysis of their structures and properties (LS_Control1_, LS_1:6_, LS_6:1_, LS_Control2_, HS_Control1_, HS_1:6_, HS_6:1_ and HS_Control2_).

### 3.2. Structural Features of CLDs

The results of the number distribution [N_de_(X)] for amylopectin (*DP* ≤ 100) and weight distribution [w(logX)] for amylose (*DP* > 100) for the eight groups of selected RS samples are shown in Figure 2 and Table 1. Except for LS_1:6_, the starch CLDs were non-selectively degraded under the shearing effect of extrusion, and the ratio of amylose to amylopectin in samples did not change significantly. In addition, as the proportion of NS increased, the proportion of short- and medium-branched chains with DP < 24 gradually increased, mainly containing degraded branched side chains and short chains with the *DB* value decreased accordingly. According to the distribution of amylose from *DP* 100 to 10,000 in Figure 2b and the data in Table 1, the proportion of amylose was substantially degraded, and the AC values of the high screw speed groups (HS_6:1_ and HS_control2_, 21.56 and 27.24%, respectively) were lower than those of the low screw speed groups (LS_6:1_ and LS_control2_, 23.13 and 29.75%, respectively), which means that stronger shearing had a greater degradation effect on the amylose in starch materials.

The CLD of amylopectin molecules and their crystal clusters (Table 1) showed that the proportion of A chains (*DP* 6–12, 11.69%) of LS_1:6_ was abnormally higher than that of the other groups (approximately 1–2%), indicating that the addition of a small amount of NS with amylose could form a large number of short branches at the low shearing level. It forms a crystal cluster with a large number of branches of the outer chains after recrystallisation, resulting in a significant increase in the *DB* to 11.06% (r = 0.918, *p* < 0.01). This high-*DB* CLD system further affected the crystal density of modified RS (*Dm* = 2.37), which in turn affected its physicochemical properties, such as viscosity (*PV*, *TV*, *FV* and *P_time_*, as described in Section 3.7). However, the rigidity of the chains in the relatively short *DP* range also prevents the proper alignment of the amylopectin double helix, relating with the number of starch crystal defects [22]. In addition, the CLD results also showed that a high proportion of A chains may lead to an increase in the proportion of RDS in the modified starch (r = 0.947, *p* < 0.01). Thus, controlling the proportion of A chains in the starch matrix (<2.5%) is essential for the efficient formation of RS.

### 3.3. Long-Range Ordering Structure

#### 3.3.1. Crystal Structure

As shown in the XRD spectra (Figure 3 and Table A1), strong diffraction peaks appeared at 2θ = 15°, 17°, 18° and 23° for LS_Control1_, LS_1:6_ and HS_Control1_, HS_1:6_, which showed an A-type crystal arrangement [30]. While the LS_6:1_, LS_Control2_ and the HS_6:1_, HS_Control2_ showed strong diffraction peaks at 2θ = 15°, 17°, 18°, 20° and 23°, presenting a C(C_b_)-type mixed crystal form [31]. That is, when A-type crystals are arranged, there is also a large number of left-handed double helices arranged in a hexagonal lattice for crystallization [32]. But none of the above changes in the crystal form of RS are correlated with the extrusion shearing strength. Meanwhile, the LS_6:1_, LS_Control2_ and the HS_6:1_, HS_Control2_ also showed V-type crystalline peaks at 2θ = 13°, which may be due to the helical complex formed by the complexation of amylose with a small amount of lipids or related compounds [33].

In addition, despite the degradation of the short-branched chains (*RC* decrease) during extrusion, the A-type crystal form of amylopectin had almost no change probably due to its high crystal density, and the process of extrusion mainly played a role in gelatinization and depolymerization with disordered chain breaking. However, for the group with a certain proportion of NS, extrusion shearing led to a significant degradation of the long amylose chains, with a large increase in the number of chains with *DP* < 24. It may have disrupted the original A-type crystal arrangement, and the rearrangement of the chains during the regrowth period had a large effect on the crystalline shape, which was manifested as a C-type mixed crystal. In this work, the relative proportion of *RC* (see Table 2) and the C6 region (see Table 3) in the starch samples was significantly negatively correlated (r = −0.884, *p* < 0.01), which further indicated that the substantial increase in side-branched chains connected by α-1,6 glycosidic bonds led to an increase in the long period of the starch lamellae after crystal rearrangement (Section 3.3.2), which led to a decrease in *RC*.

#### 3.3.2. Long-Period/Lamellar Distance Structure

SAXS was used to detect the interlayer distance (*d_Bragg_* and *d_Lorentz_*) and the homogeneity (*Dm*) of the starch crystal arrangement, as shown in Figure 4. All groups of high-RS samples showed a new lamellar arrangement after extrusion and retrogradation, which was characterized by *d_Lorentz_* (“shoulder-like” peaks in Figure 4a). And the period range was from 17.35 to 22.07 nm, which was significantly higher than that of the original *d_Bragg_*. This scattering peak pattern was consistent with the experimental phenomenon of Zhang et al. [34]. Table 2 shows that the long-period structure of extrusion-modified RS was hardly influenced by the shearing levels, while it was correlated with the original CLDs of starch. The *d_Bragg_* values (8.83–9.59 nm) of extruded starch were retained in the high-amylopectin samples (with A-type crystals), and their *d_Lorentz_* values were lower than those of the samples with a certain proportion of amylose (LS/HS_6:1_ and LS/HS_Control2_). It suggested that the relatively short-branched chains of starch (with *DP* < 24) were mainly attached to the branched chains of APC (e.g., B2, B3 chains) during retrogradation, rather than the extension of the C chain as the backbone. It is conjectured that the B3 chains (*DP* 37–100) might be attached to the C chain of the APC unit, with a negative correlation with *d_Lorentz_* (r = −0.795, *p* < 0.05). In addition, the *d_Lorentz_* value of starch lamellae gradually increased with increasing amylose content by adding NS.

*α* (*Dm* = *α*, if 1 < *α* < 3) can be calculated and used to describe the starch mass fractal structure, and the density of the scattered starch sample increases with an increase in *Dm* value. In general, when the starch crystalline lamellae are uniform in size and spacing, the scattering peaks in Figure 4b show a high, narrow peak type [35]. The results showed that the crystal arrangements of extrusion-modified RS were well homogeneous, with *Dm* values ranging from 2.02 to 2.37. *DB* was significantly and positively correlated with *Dm* (r = 0.871, *p* < 0.01), which means that high-branching CLDs were beneficial to the enhancement of the crystal densities.

### 3.4. Short-Range Ordering Structure

The short-range ordering structure of extrusion-modified RS can be effectively analyzed by the indicators of *DO*, *DD* and the relative proportions of the bonds at C1, C4 and C6 obtained by FTIR and ^13^C NMR. The FTIR spectra of extruded starch (Figure 5) reflected the changes in short-range molecular structure, such as double helix structure and chain conformation [36]. They are divided into three regions: 800–1500 cm^−1^ (fingerprint region), 2800–3000 cm^−1^ (C-H stretching region) and 3000–3500 cm^−1^ (O-H stretching region). Among them, the broad peaks in the range of 3000–3500 cm^−1^ are related to the free, intermolecular and intramolecular hydrogen bonding stretching vibrations [37,38]. While the absorption peak near 2929 cm^−1^ can be attributed to the asymmetric stretching of C-H, the absorption near 1648 cm^−1^ is due to the presence of bound water in the starch [39]. By deconvolution, the FTIR results with short-range ordering information are shown in Figure 5b. According to the peak intensities at 1047cm^−1^, 1022cm^−1^ and 995cm^−1^, the degree of order R_1047/1022_ (*DO*) and the double helicity R_995/1022_ (*DD*) were calculated, and the data are listed in Table 3. *DO* represents the ratio of starch crystalline area to amorphous area [40], while *DD* not only reflects the double helix structure in the crystal structure but also reflects the short-range molecular ordering of the amorphous region [41]. The *DD* formed by the rearrangement of starch samples was decreased by extrusion and retrogradation, while the *DO* value seldom changed (compared to native starch). It is considered to be a consistent result of the orderly structure of starch chains to form new *d_Lorentz_*, as described in Section 3.3.2. In addition, we found that *DP* 12–24 is the range greatly connected with newly formed long-period *d_Lorentz_* after extrusion. The crystal region and amorphous region (as *DO* value) of high-RS samples show a similar periodic arrangement of length, and the crystal structures are uniform, which is consistent with the results of the SAXS analysis mentioned above.

All high-RS sample groups showed similar ^13^C NMR patterns, indicating that extrusion shearing did not promote the formation of new functional groups of starch. This mechanical process only involved the degradation of starch chain and hydrogen bond rearrangement, that is, the prepared RS belongs to physically modified RS3. For the APC growth model of RS, it is speculated that its association modes can be non-covalent bonding and helix and spatial cross-linking. The relative proportions of bonds at C1, C4 and C6 detected by ^13^C NMR (Table 3) can be used to comprehensively analyze the chain dissociation and bond formation of starch chains at α-1,4 and α-1, 6 glycosidic bonds (as described in Section 3.3.1).

### 3.5. Microscopic Morphology of Extruded Starch

Four groups of typical extruded starch samples were selected (with similar RS content) in order to observe if they had a difference in their morphologies (Figure 6a–d). They included the high-RS samples with different structures based on CLDs, as well as other molecular structure information. Figure 6a shows the “melting” morphology of the high-amylopectin group (LS_control1_), whereas in LS_1:6_ with the addition of a small amount of amylose in NS, the amylopectin branches seemed to be reconnected by the amylose skeleton at the retrogradation stage, extending to form a “honeycomb-like” porous micromorphology, as shown in Figure 6b. At high shearing levels of the extrusion process, HS_control2_ showed the “filamentous” streak pattern (Figure 6d), and by decreasing the content of amylose in the group of HS_6:1_, the layer of arranged crystals changes to be thickened, probably due to the high thermomechanical force during extrusion and increased amylopectin during retrogradation, presenting the “lamellar” microscopic pattern shown in Figure 6c. This result was considered to be partly consistent with the difference in long/short-range ordering structure described in Section 3.3 and Section 3.4.

### 3.6. Thermal Properties

The thermal properties of high-RS samples were analyzed, and the data are shown in Table 4. Starch granules usually show a single melting peak of gelatinization, but in this study, RS samples showed a melting of two crystal structures in the DSC curve, which is consistent with the findings of Waigh et al. [26]. They noted that there are two stages in the starch gelatinization process: the first is the slow isotropic transformation of aligned chains or amylopectin side chains, followed by the breaking of thermostable microcrystals. Since the 1980s, several models have been proposed to explain the heat-absorption transition of starch–water biphasic systems, such as expansion-driven melting [42] and microcrystalline stability [43]. However, these models seem to neglect the structural changes in starch crystals during the heat process. The samples of high RS have different original CLDs, which induced specific interchain interactions during the DSC test [44], causing secondary structural transformations in the starch crystals.

The first heat absorption peak of high-RS samples was around 51.68–57.16 °C, and Δ*H*_1_ was positively correlated with RS content (r = 0.745, *p* < 0.05) and negatively correlated with RDS content (r = −0.736, *p* < 0.05). It is illustrated that Δ*H* reflects the energy required to destroy the helical structure during starch gelatinization [45]. Thus, the phenomenon of RS increase is inferred to be related to the disruption or rearrangement of hydrogen bonds within the helix in the crystalline zone. Meanwhile, *T*_*p*1_ showed a positive correlation with the content of A chains (r = 0.835, *p* < 0.01), *DB* (r = 0.829, *p* < 0.01) and *Dm* (r = 0.794, *p* < 0.05), which suggests that the formation of the first peak of heat adsorption is probably related to the melting of the helical structure of branched clusters based on the A chains. The second heat absorption peak occurred around 77.52–100.29 °C and is generally considered to be in the range of the gelatinization peaks exhibited by natural crystals of starch [6]. In this study, the second gelatinization peaks were detected only in samples of LS_Control1_, LS_Control2_, HS_Control1_ and HS_1:6_, and the Δ*H_2_* values were relatively low. It is assumed that the rearrangement of the crystalline structure is dominant in the extrusion-modified RS, which resulted in the disruption of the original crystal structure. In addition, the Δ*H*_1_ values of LS_Control1_ and LS_Control2_ were relatively high, and these pure starch systems of CLDs at the low shearing level were more favorable than those at the high shearing level to improve the thermal stability of the rearranged crystals in extruded starch.

### 3.7. Viscosity Characteristics

The viscosity property of starch can be evaluated by RVA, and the results are presented in and Table 5. *PV*, *TV* and *FV* indicators were all significantly and positively correlated with each other, which reflects the swelling, disintegration and pasting ability of starch samples [46]. Comparing different original CLDs at the same screw speeds, it was found that the *AC* of extrusion-modified RS was positively correlated with the changes in *PV* (r = 0.768, *p* < 0.05), *FV* (r = 0.808, *p* < 0.05) and *TV* (r = 0.747, *p* < 0.05), which was consistent with the results of Li et al. [27]. The samples with a certain proportion of straight-chain starch (≥27.97%) were more likely to develop high viscosity. It is speculated that when starch granules are heated in an aqueous solution which causes the granules to rupture, the leaching of amylose will further promote the swelling of the granules, thereby increasing the viscosity. *BD* can be used to measure the degree of starch disintegration [47]. *SB* reflects the viscosity properties of gelatinized starch at the stage of cooling and regeneration to form gel, and the increase in this value is related to the rearrangement of amylose molecules [48]. LS_6:1_ and HS_6:1_ have a higher amylopectin content than the LS_Control2_ and HS_Control2_, respectively, which facilitates rapid cross-linking and shaping during retrogradation, and therefore has higher *SB* values. In addition, the *P_time_* of the samples also reflected the pasting stability of starch, with the lowest value of 6.62 min in the LS_1:6_ group and the highest value of 12.80 min in LS_6:1_. The *P_time_* was analyzed to be negatively correlated with the A-chain content (r = −0.712, *p* < 0.05) and *DB* value (r = −0.777, *p* < 0.05) but positively correlated with *RC* (r = 0.800, *p* < 0.05). It indicates that the expanded growth of the APC of the LS_1:6_ group was beneficial to the improvement of starch crystal densification and thermal stability (as described in Section 3.3.2 and Section 3.6) but still reduced the gelatinization stability of starch. This is presumably related to its larger long-period arrangement (*d_Lorentz_*), which has a larger spatial resistance in solution and is less prone to aggregation [49].

## 4. Conclusions

In this work, starch products with high RS3 content were prepared from different original CLDs by extrusion at low/high levels. The effects of CLD on the multiscale structure and physicochemical properties of RS were investigated. A high RS content was observed at a high extrusion shearing level based on high-amylopectin starch materials. By the extrusion of APC in starch, B3 chains were linked to the C chain of amylopectin; B1 chains, A chains and ultra-short chains (*DP* < 6) obtained by screw shearing were probably connected outward to APC. By retrogradation, crystal clusters in extruded starch with a large number of branched chains were formed. The crystalline and amorphous regions of extruded starch with a high RS content had similar and homogeneous crystals but different modified CLDs, long/short-length ordering structures and morphologies. This study clarified that CLD is a key factor influencing the formation and arrangement of RS3 fine structures formed by extrusion, which can provide a reference for regulating starch fine structures by extrusion and designing RS3 processing strategies with target structures and performance.

## Figures and Tables

**Figure 1 foods-13-02532-f001:**
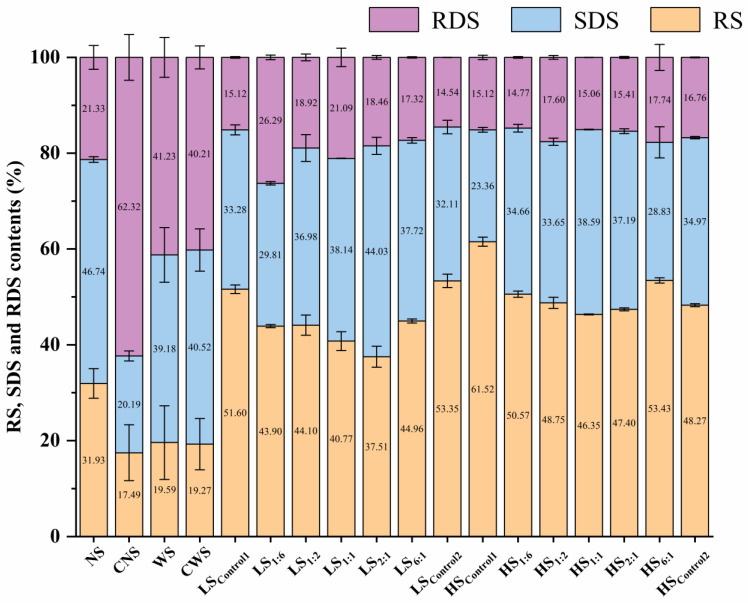
In vitro digestion (RS, SDS and RDS contents) of starch samples. NS: raw normal maize starch; CNS: cooked normal maize starch; WS: raw waxy maize starch; CWS: cooked waxy maize starch; Control1: waxy starch, WS; Control2: normal starch, NS; 1:6–6:1: ratio of NS to WS; LS: 50 r/min as low screw speed; HS: 250 r/min as high screw speed.

**Figure 2 foods-13-02532-f002:**
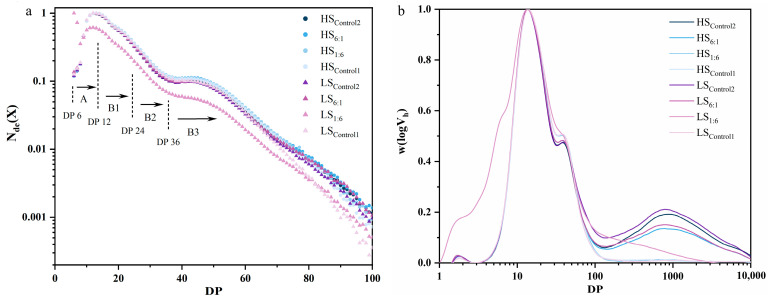
The chain-length distributions of amylopectin (**a**) and amylose (**b**) in extruded starch. Control1: waxy starch, WS; Control2: normal starch, NS; 1:6–6:1: ratio of NS to WS; LS: 50 r/min as low screw speed; HS: 250 r/min as high screw speed.

**Figure 3 foods-13-02532-f003:**
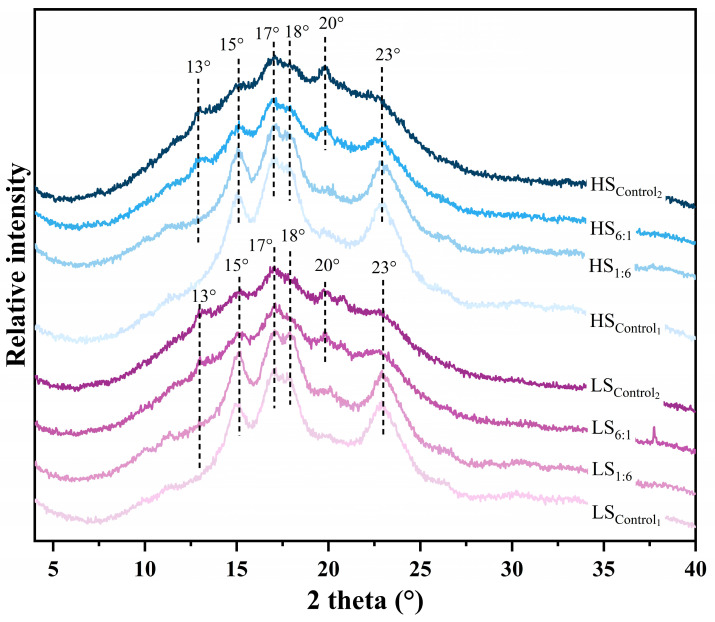
X-ray diffraction (XRD) patterns of extruded starch. Control1: waxy starch, WS; Control2: normal starch, NS; 1:6–6:1: ratio of NS to WS; LS: 50 r/min as low screw speed; HS: 250 r/min as high screw speed.

**Figure 4 foods-13-02532-f004:**
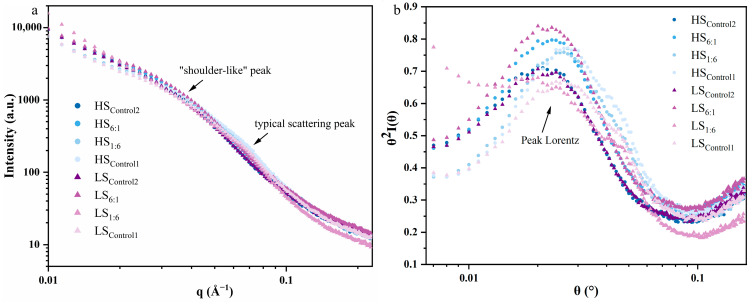
Small-angle X-ray scattering intensity profiles of Bragg (**a**) and Lorentz (**b**) of extruded starch. Control1: waxy starch, WS; Control2: normal starch, NS; 1:6–6:1: ratio of NS to WS; LS: 50 r/min as low screw speed; HS: 250 r/min as high screw speed.

**Figure 5 foods-13-02532-f005:**
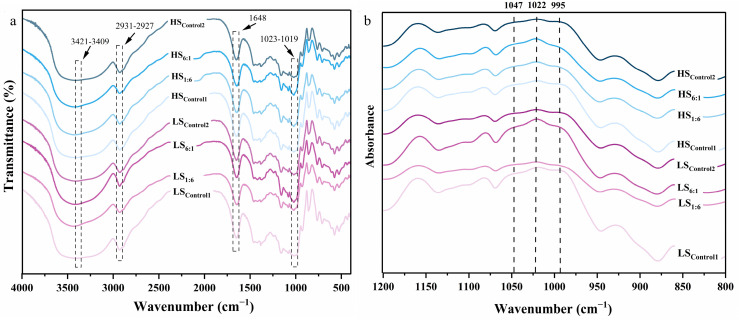
Fourier transform infrared spectra (**a**) and their deconvoluted spectra (**b**) of extruded starch. Control1: waxy starch, WS; Control2: normal starch, NS; 1:6–6:1: ratio of NS to WS; LS: 50 r/min as low screw speed; HS: 250 r/min as high screw speed.

**Figure 6 foods-13-02532-f006:**
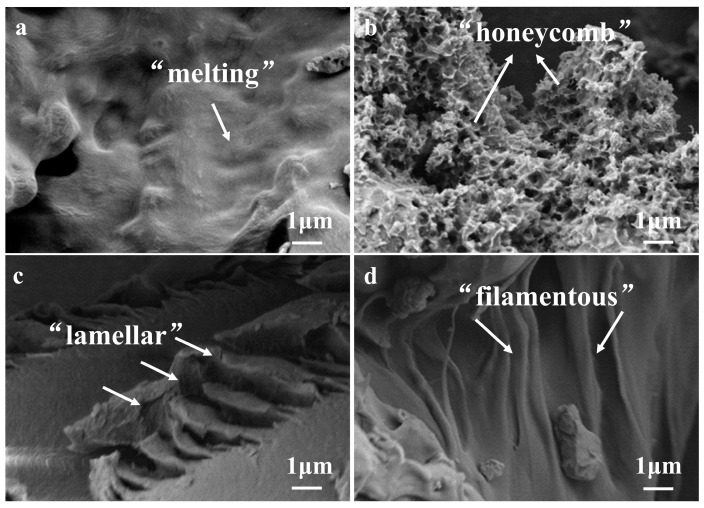
Scanning electron microscope images of extruded starch: (**a**) LS_Control1_; (**b**) LS_1:6_; (**c**) HS_6:1_; (**d**) HS_Control2_.

**Table 1 foods-13-02532-t001:** Molecular parameters of amylopectin and crystal clusters of extruded starch.

Number	DP < 6	DP 6–12	DP 13–24	DP 25–36	DP 37–100	AC (%)	DB (%)
LS_Control1_	14.21	1.42	41.59	26.24	16.55	2.74	6.32
LS_1:6_	17.92	11.69	39.88	18.38	12.05	7.96	11.06
LS_6:1_	25.43	1.88	39.34	19.41	13.83	23.13	5.20
LS_Control2_	20.04	2.18	44.51	19.44	13.75	29.75	4.47
HS_Control1_	19.87	2.21	44.38	19.20	14.30	3.00	6.32
HS_1:6_	19.94	1.48	37.56	24.34	16.56	2.77	6.38
HS_6:1_	25.57	1.84	37.53	20.35	14.60	21.56	5.26
HS_Control2_	22.91	2.00	42.65	18.77	13.60	27.24	4.90

DP, AC and DB are the degree of polymerization, content of amylose and degree of branching, respectively.

**Table 2 foods-13-02532-t002:** The relative crystallinity and long-period distances (length of lamellae) of extruded starch.

Number	RC (%)	Dm	q (Å^−1^)	θ (°)	d_Bragg_ (nm)	d_Lorentz_ (nm)
LS_Control1_	51.61	2.04	0.066	0.026	9.59	16.97
LS_1:6_	46.26	2.37	0.070	0.020	9.01	22.07
LS_6:1_	60.58	2.05	-	0.022	-	20.06
LS_Control2_	61.49	2.07	-	0.022	-	20.06
HS_Control1_	50.69	2.03	0.070	0.027	9.01	16.35
HS_1:6_	54.97	2.02	0.071	0.025	8.83	17.65
HS_6:1_	46.40	2.05	-	0.023	-	19.19
HS_Control2_	58.50	2.08	-	0.021	-	21.01

RC and Dm are the relative crystallinity and fractal dimension, respectively.

**Table 3 foods-13-02532-t003:** The analysis of the short-range orders of extruded starch.

Number	DO	DD	C1 (%)	C4 (%)	C6 (%)
LS_Control1_	0.9367	0.9733	18.24	5.85	18.61
LS_1:6_	0.9381	0.9221	15.75	5.70	19.04
LS_6:1_	0.8945	0.9012	16.54	5.85	16.70
LS_Control2_	0.9399	0.9557	22.11	6.75	17.41
HS_Control1_	0.9449	0.9571	15.87	6.42	20.10
HS_1:6_	0.9224	0.9149	14.83	5.45	18.45
HS_6:1_	0.9084	0.8587	16.28	7.82	20.15
HS_Control2_	0.9541	0.9623	16.53	5.72	17.14

DO and DD are the degree of ordering and degree of double helix, respectively.

**Table 4 foods-13-02532-t004:** The analysis of the thermostability of extruded starch.

Number	T_o_ (°C)	T_p1_ (°C)	T_p2_ (°C)	T_c_ (°C)	ΔH_1_ (J/g)	ΔH_2_ (J/g)
LS_Control1_	46.90 ± 2.39 ^ab^	52.02 ± 3.48 ^b^	78.73 ± 0.23	83.60 ± 0.20 ^b^	2.93 ± 1.94 ^a^	1.22 ± 0.43
LS_1:6_	50.24 ± 1.57 ^a^	57.16 ± 0.82 ^a^	-	64.01 ± 3.08 ^c^	0.42 ± 0.30 ^b^	-
LS_6:1_	46.89 ± 1.86 ^ab^	51.68 ± 1.41 ^b^	-	58.55 ± 1.80 ^c^	0.84 ± 0.55 ^ab^	-
LS_Control2_	47.30 ± 1.68 ^b^	52.84 ± 0.58 ^b^	100.29 ± 0.60	104.33 ± 0.02 ^a^	2.68 ± 1.70 ^a^	0.92 ± 0.45
HS_Control1_	45.71 ± 1.99 ^b^	53.44 ± 0.27 ^b^	77.52 ± 0.23	82.96 ± 1.39 ^b^	2.65 ± 1.16 ^a^	0.76 ± 0.51
HS_1:6_	48.64 ± 0.90 ^ab^	54.38 ± 0.69 ^b^	78.09 ± 0.26	85.24 ± 1.51 ^b^	1.68 ± 0.28 ^ab^	1.14 ± 0.56
HS_6:1_	47.47 ± 1.17 ^ab^	54.17 ± 0.61 ^b^	-	61.31 ± 2.48 ^cd^	1.19 ± 0.59 ^ab^	-
HS_Control2_	46.40 ± 1.38 ^ab^	52.92 ± 0.49 ^b^	-	59.53 ± 2.14 ^d^	1.18 ± 0.52 ^ab^	-

Different letters represent significant differences (*p* < 0.05) in the superscripts of the same column.

**Table 5 foods-13-02532-t005:** Pasting properties of extruded starch.

Number	PV (cP)	TV (cP)	FV (cP)	BD (cP)	SB (cP)	P_time_ (min)
LS_Control1_	481 ± 4 ^d^	369 ± 5 ^d^	495 ± 3 ^e^	112 ± 2 ^a^	126 ± 2 ^c^	8.11 ± 0.03 ^f^
LS_1:6_	23 ± 1 ^h^	15 ± 1 ^g^	21 ± 1 ^g^	8 ± 0 ^ef^	6 ± 0 ^f^	6.62 ± 0.03 ^g^
LS_6:1_	650 ± 4 ^b^	649 ± 4 ^b^	900 ± 7 ^a^	2 ± 1 ^h^	252 ± 4 ^a^	12.80 ± 0.20 ^a^
LS_Control2_	595 ± 5 ^c^	559 ± 4 ^c^	621 ± 4 ^c^	36 ± 2 ^c^	62 ± 1 ^e^	11.22 ± 0.11 ^c^
HS_Control1_	199 ± 1 ^f^	188 ± 1 ^e^	251 ± 2 ^f^	11 ± 1 ^e^	63 ± 1 ^e^	10.62 ± 0.08 ^d^
HS_1:6_	174 ± 1 ^g^	171 ± 1 ^f^	237 ± 1 ^f^	3 ± 0 ^fg^	66 ± 1 ^e^	11.84 ± 0.41 ^b^
HS_6:1_	433 ± 2 ^e^	382 ± 4 ^d^	513 ± 3 ^d^	52 ± 3 ^b^	131 ± 2 ^b^	9.36 ± 0.28 ^e^
HS_Control2_	736 ± 18 ^a^	713 ± 16 ^a^	786 ± 19 ^b^	23 ± 6 ^d^	73 ± 3 ^d^	11.96 ± 0.08 ^b^

Different letters represent significant differences (*p* < 0.05) in the superscripts of the same column. PV, TV, FV, BD, SB and P_time_ are the peak viscosity, trough viscosity, final viscosity, breakdown value, setback value and pasting time, respectively.

## Data Availability

The original contributions presented in the study are included in the article, further inquiries can be directed to the corresponding author.

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
