# Peer review of "High-Resistant Starch Based on Amylopectin Cluster via Extrusion: From the Perspective of Chain-Length Distribution and Structural Formation"

_foods, 2024, doi:10.3390/foods13162532_

Round 1

Reviewer 1 Report

Comments and Suggestions for Authors

The manuscript “High Resistant Starch Based on Amylopectin Cluster via Extrusion: From the Perspective of Chain-Length Distribution and Structural Formation” is interesting and provides a valuable, comprehensive, and well-executed characterization of starch through various approaches. However, these interesting results are undermined by poor discussion and insufficient explanations of the behavior observed. It is recommended that the authors enhance their discussion and compare the obtained data with results available in the literature. Furthermore, it is suggested that the authors provide a mechanism behind the increase in resistant starch. Finally, some specific comments for the authors are provided:

1. Line 36: “RS is the anti-digestion structure of pure starch” – What does this mean? Does it imply that starch is fully digestible in its native state (with no treatments applied)? Could you explain the differences between RS and native starch, noting that RS2 is known as a native starch (e.g., Hylon VII with 70% amylose)?

2. Line 2: “too low or too high ratio of A to B chains” – When is it considered too high or too low? It would be more substantial if the authors provided a specific number or ratio range.

3. Introduction: Please add some lines in the introduction section providing more evidence of studies on high-solid extrusion and simple extrusion processing focused on promoting RS formation.

4. Line 74: Please provide the purity of waxy maize starch.

5. Lines 84-85: Consider using “rpm” instead of “r/min.”

6. Lines 87-88: Could the milling process affect the RS3 amount? Please specify the grinding method used and the conditions.

7. Line 92: Please provide the mesh size used. It is interesting that in lines 87-88, the authors mentioned that the starch sample was ground, but in line 94 it is explained that 20 to 10 mesh was used to avoid mechanical damage to the starch crystal structure. These statements are contradictory. Therefore, it is important to clearly describe the procedure in lines 87-88 for grinding the sample and how the authors ensured no mechanical damage occurred during this step.

8. Lines 107-110: Please explain the meaning of each acronym used and the equations (1-3).

9. Section 2.4.1: Please provide information about the HPLC equipment used (model, brand, and manufacturer country). Additionally, add information about the chromatographic method: mobile phase, temperature, and injection amount. Also, provide information about the standards used to estimate the DP.

10. Line 139: Please add the mesh size in microns and state if it was a standardized US mesh. For example: “(sieved through a 200 U.S. mesh, 74 μm).”

11. Line 139: Please provide information about the diffractometer used (brand, model, and manufacturer country).

12. Section 2.5.1: Lines 139 to 142 contain a sentence that is too long. Please consider rewriting it and splitting it into two sentences. Suggested text structure: “XRD was used to analyze the crystal form and relative crystallinity (RC) of starch. Fifty milligrams of ground and sieved sample were scanned using a diffractometer (model, country of manufacturer), operated at 40 kV and 40 mA, using Cu-Kα radiation (λ = 0.1542 nm). The sample was scanned from 4° to 40° (2θ scale) with a step size of 0.02°. The diffraction peaks (...)”

13. Section 2.8: Provide information about the SEM equipment (brand, model, and manufacturer country).

14. Section 2.10: Provide information about the RVA equipment (brand, model, and manufacturer country).

15. Line 210: The term “anti-digestibility” sounds odd. Consider using "resistant starch" (RS) directly.

16. Line 216: The authors mentioned that melting of starch may occur during double screw extrusion. According to the literature, can starch melt at certain temperatures? If so, what is the melting temperature of starch, and what occurs structurally during “starch melting”?

17. Figure 1: Please write the term “in vitro” in italics in the figure caption.

18. What are the RDS, SDS, and RS content in raw waxy and normal starches? It is highly recommended that the authors provide this information.

19. Lines 227-232: How could amylopectin be transformed into amylose? The writing is a bit confusing. Please revise and support this statement.

20. Line 421: The authors state that as the proportion of NS increased, the content of chains (DP<24) gradually increased. However, this behavior is not clearly noticeable in Table 1.

21. Line 274: The authors mentioned strong diffraction peaks at 15°, 17°, 18°, 20°, and 23° (2θ scale). However, these peaks are part of the orthorhombic structure of starch representing the hkl directions of 121 (15.08°), 031 (17.3°), 211 (18.09°), 040 (20.183°), and 231 (23.029°), respectively. Therefore, no structural change was observed, only changes in intensity, which is consistent with the relative crystallinity displayed in Table 2. It is recommended to calculate the second derivative of their XRD data to observe significant changes in diffraction peaks (new peaks or angular shifts). Particularly, HScontrol2 and HS6:1 and their respective LS treatments show a change in their crystalline structure due to the significant increase in the signal at peak 20°. Moreover, these treatments presented a new peak at 13°, indicating a structural change of starch from A to a V-type pattern. Please revise.

22. Table 3: Order degree and degree of double helices of starch is usually reported in values around 0.2 to 6. How do the authors explain their treatment having an OD and DD >85?

23. Line 349: Please provide all the characterizations conducted on native starches (waxy and NS).

24. SEM: How can the authors assure that melting is occurring instead of gelatinization? Note that previous studies have reported dry gelatinization (moisture <20%).

25. Thermal properties: Why did the authors analyze temperatures <100 °C? Typically, the region between 30 and 160 °C is explored to evaluate the thermal properties of resistant starch (RS3 or RS5), showing thermal events at approximately 115°, 120°, and 130 °C.

26. Line 419: Consider using “evaluated” instead of “detected”; consider removing the citation [41] as it is not necessary in this line.

27. Line 420: When first mentioning a property in the text, it is recommended to use the full term and then use the abbreviation. For example: “Peak viscosity (PV).”

28. Figure 7: The viscosity profile in some treatments showed two peaks (e.g., LS6:1). Please examine this behavior and provide a well-supported explanation. Consider that the DP has changed due to the extrusion process, influencing interaction dynamics and the ability to resist flowing. Additionally, note that these two peaks develop in different temperature ranges, suggesting two main solubility behaviors: at cold temperatures (above 40 °C) and high temperatures (near 80°C). This may indicate the presence of both gelatinized and ungelatinized starch in the sample (as XRD indicates). Please revise and complete this section.

29. The terms “correlated” and “correlation” are frequently used throughout the manuscript, but the correlation analysis and coefficients are not shown. Did the authors conduct a correlation analysis, or are they using the term loosely? Please revise and clarify.

Reviewer 2 Report

Comments and Suggestions for Authors

The manuscript investigates the effect of extrusion of corn starch with different amylose content on formation of resistant starch and its structure. The experiment design ins correct and the results are clearly presented. The manuscript is prepared quite thoroughly, with one exception – the introduction section, which seems chaotic. The authors tough only the surface of the problem, while the reason underlying the undertaken topic is not quite clear (based on the provided information).

Minor remarks:

The data in tables would benefit form two-way analysis of variance considering factors: amylose content and screw speed

Line 36 this definition of RS3 is “strange”

Line 455 -this conclusion cannot be extrapolated to all RS3 preparation, but only to investigated case and presumably to other extruded starches it should be rephrased

The literature is properly selected with board timeline, but few more recent studies could be considered as well.

Reviewer 3 Report

Comments and Suggestions for Authors

This study investigated the effect of the ratio of normal corn starch and extrusion conditions on the structural properties and digestibility of retrograded waxy corn starch. The physicochemical properties of the starch samples were effectively evaluated using well-defined analytical techniques. The authors also discussed how the structural properties of the samples impact the physicochemical and digestive properties of the starch samples. I suggest that the authors consider the points addressed below:

  • Please add results of retrograded, cooked native normal starch and waxy starch (cooked in a boiling water bath) without extrusion cooking.
  • Pearson correlation may be helpful to evaluate the relationships between data.
  • Molecular structure (L257-259): Why does the addition of a small amount of NS increase DP6-12?
  • Debranched starch is often used to evaluate the fine amylopectin molecular structure. In this study, the fine amylopectin molecular structure is also an important consideration. However, since intact molecules were retrograded, please show the intact molecular structure of the samples without the debranching process.
  • XRD: It appears that some amylose inclusion complexes were generated during the process. Please discuss this.
  • Viscosity: Please explain the RVA results in more detail. For example, why did LS6:1 show very low viscosity?
Comments on the Quality of English Language

N.A.

Reviewer 4 Report

Comments and Suggestions for Authors

The manuscript describes the production of resistant starch with normal and waxy corn starch via extrusion. There are critical flaws in experimental design and discussion. A further consideration of publication could proceed after addressing the following critical questions.

1.     Starch treated via conventional thermal processing should be control group in this study to support the effect of extrusion.

2.     CLD of extruded starch or extruded/retrograded starch was analyzed in this study? Specify the sample. The author stated “8 groups of selected RS samples” in line 237.

3.     In case of normal starch, high screw speed induced lower RS (53.35% 48.27%). Moreover, the mixture of normal and waxy starch showed lower RS content than normal and waxy alone. Abstract (Line 24_“high amylopectin content and high screw speed of extrusion was favorable for RS3 formation”) should be revised according to data. Discussion about RS content should be added in section 3.1. 

Comments on the Quality of English Language

Moderate editing of English language required

Round 2

Reviewer 3 Report

Comments and Suggestions for Authors

Manuscript was properly revised according to suggestion of reviewers.

Reviewer 4 Report

Comments and Suggestions for Authors

The authors have addressed my comments. I will suggest the acceptance of this work.